# Reduced representation approaches produce similar results to whole genome sequencing for some common phylogeographic analyses

Drew J. Duckett[1], Kailee Calder[2], Jack Sullivan[3], David C. Tank[4], Bryan C. Carstens[1]*

**1** Department of Evolution, Ecology, and Organismal Biology, The Ohio State University, Columbus, OH, United States of America, **2** College of Veterinary Medicine and Biomedical Sciences, Colorado State University, Fort Collins, CO, United States of America, **3** Department of Biological Sciences, University of Idaho, Moscow, ID, United States of America, **4** Department of Botany, University of Wyoming, Laramie, WY, United States of America

* carstens.12@osu.edu

**Data Availability Statement:** Raw sequencing reads for both genotyping by sequencing will be submitted to the Sequence Read Archive (SRA),

## Abstract

When designing phylogeographic investigations researchers can choose to collect many different types of molecular markers, including mitochondrial genes or genomes, SNPs from reduced representation protocols, large sequence capture data sets, and even whole genomes. Given that the statistical power and accuracy of various analyses are expected to differ depending on both the type of marker and the amount of data collected, an exploration of the variance across methodological results as a function of marker type should provide valuable information to researchers. Here we collect mitochondrial Cytochrome *b* sequences, whole mitochondrial genomes, single nucleotide polymorphisms (SNP)s isolated using a genotype by sequencing (GBS) protocol, sequences from ultraconserved elements, and low-coverage nuclear genomes from the North American water vole (*Microtus richardsoni*). We estimate genetic distances, population genetic structure, and historical demography using data from each of these datasets and compare the results across markers. As anticipated, the results exhibit differences across marker types, particularly in terms of the resolution offered by different analyses. A cost-benefit analysis indicates that SNPs collected using a GBS protocol are the most cost-effective molecular marker, with inferences that mirror those collected from the whole genome data at a fraction of the cost per sample.

## Introduction

The introduction of high throughput technologies for DNA sequencing has enabled new approaches for generating the molecular data used to investigate population and species histories [1]. Methods such as restriction associated digest sequencing (RADSeq; [2]) and hybridization capture [3] make it feasible to collect molecular markers that are representative of the entire nuclear genome at a modest financial cost. The increasing affordability of high throughput sequencing also means that population level data sets based on whole nuclear genomes are

and intermediate coverage whole genome sequencing reads are available in SRA (PRJNA809709). Whole mitochondrial genome sequences have been deposited in GenBank (MT225016.1, MW387162, MW387163, OM501052-OM501078). All custom scripts are available from GitHub (github.com/djlduckett/Genome_Resources). All VCF files, alignments, data analysis files, phylogenetic trees, and custom software are available from Zenodo (https://zenodo.org/records/10059138).

**Funding:** Directorate for Biological Sciences, DEB-1457519, Dr Bryan Carstens Ohio Supercomputer Center, PAS1431, Dr Bryan Carstens Ohio Supercomputer Center, PAA0202, Dr Bryan Carstens National Museum of American History, Theodore Roosevelt Memorial Grant, Dr. Drew J. Duckett Society of Systematic Biologists, Graduate Student Research Grant, Dr. Drew J. Duckett The Ohio State University (Graduate Fellowship) to DJD.

**Competing interests:** The authors have declared that no competing interests exist.

becoming more common in molecular ecology (e.g., [4, 5]). As a result, over the last two decades phylogeographic investigations have increased from using data from a single mito-chondrial locus to a handful of microsatellites or nuclear sequences to tens of thousands of single nucleotide polymorphisms (SNPs) (e.g., [6, 7]). While phylogeographic investigations with 10,000+ SNPs are now common, it is unclear whether the costs associated with collecting large SNP data sets are justified and at what point it would be worthwhile to collect whole genome data.

Genes from organellar genomes were once the most commonly used markers in phylogeography because they were easy to sequence using Sanger methods. Some 20 years ago, the discipline began to question the utility of such markers that are for the most part non-recombining and typically reflect only maternal history (reviewed by [8]). Researchers began utilizing nuclear sequence markers (e.g., [9]) and quickly pivoted to SNPs (e.g., [10, 11]) and sequence capture probe data (e.g., [12, 13]). Throughout this transition, researchers have generally considered larger datasets to be more desirable because they were assumed to contain more informative. This intuition likely results from simulation studies that were designed to show that estimates made from a single locus were likely to be inadequate (e.g., [14, 15]), rather than a comprehensive exploration of results across different markers. We can potentially collect *all* of the sequence data from a particular taxon, but it is far from clear which proportion of these data are required to address common research questions. Many phylogeographic investigations seek to infer the number of populations present in a species (i.e., population genetic structure), estimate changes in historical population sizes, and understand the relationships among populations. In these scenarios, it would be useful to know how well estimates derived from different marker types approximate those from whole genome sequencing, which presumably represents the upper limit on the amount of available data. A comparative evaluation across marker types would provide guidance that could help researchers to allocate their finite financial resources in an effective manner.

The choice of molecular marker can affect the power and accuracy of analyses but will also be influenced by other factors. For example, sequencing protocols vary in the amount, quality, and concentration of input DNA required to obtain good results, for example restriction-digest based methods [2] generally require lower concentrations than sequence capture probe methods such as ultraconserved elements (UCEs), while UCEs can be successfully sequenced from lower quality input DNA [3]. Particularly for phylogeographic investigations into rare or threatened focal species, the availability of tissues could circumscribe researcher choices about molecular markers. While marker choice may be influenced by sample availability, each sequencing method has possible shortcomings that should also be considered. Sequence library generation approaches including RADSeq or Genotyping by Sequencing (GBS) can be affected by factors such as allelic dropout and rely on locus clustering thresholds in the absence of a closely related reference genome [16–18]. Capture probe techniques such as UCEs do not represent a random sample of the focal genome, and therefore may introduce bias [3].

Recent investigations have compared high-throughput sequencing datasets. For example, Smith et al. [19] compared GBS and whole genome sequencing (WGS) to examine demographic models with approximate Bayesian computation. They found largely consistent results between the two marker types, despite some differences in parameter estimates. In contrast, Duntsch et al. [20] found that WGS provided better estimated runs of homozygosity compared to SNP arrays and RADSeq, although SNP density and depth had a large effect. Similarly, Szarmach et al. [21] recently showed that WGS performed better than GBS and RADSeq when examining divergence landscapes. While these studies might imply that WGS is worth the extra cost, they did not examine many common analyses across a range of marker types. Specifically, analyses of population genetic structure, changes in historical demography, and

genetic distance among samples are critical to many phylogeographic investigations and would benefit from such a comparison.

Our investigation explores these issues by comparing different results from multiple methods that are commonly used in phylogeographic analysis across marker types. Specifically, we separately collected data from GBS and WGS and extract whole mitochondrial genomes (mtgenomes), Cytochrome *b* (cytb), and UCE sequences from the WGS data. We then analyze each data set using several analytical approaches: analysis of molecular variance (AMOVA), genetic distance, principal component analysis (PCA), ADMIXTURE analysis, Bayesian Skyline plots, and Stairway plots. We used samples from the North American water vole (*Microtus richardsoni*), a species with a recent evolutionary history dominated by Pleistocene glaciation and post-glacial expansion [22], as an empirical example because we wanted to ensure that the results of this study would reflect the conditions that other researchers would be likely to encounter.

## Materials and methods

### Study system

The North American water vole, *Microtus richardsoni* [23], has historically been given a number of names by taxonomists, including *Arvicola riparius* [23], *Arvicola richardsoni* [24], and *Aulacomys arvicoloides* [25]. Most recently and coincident with arguments in favor of elevating the subgenus *Mynomes* to generic status, the species has recently been renamed *Mynomes richardsoni* [26]. We refer to this taxon as *Microtus richardsoni* here to provide continuity with previous work during this ongoing taxonomic debate. *M. richardsoni* consists of four recognized subspecies in the Pacific Northwest of North America (Fig 3; [27]). *M. r. arvicoloides* [28] occupies the Cascades Mountains, from southern Canada to southern Oregon and includes the synonym *principalis* described by Rhoads [28]. No variants of *M. richardsoni* are found in the Columbia Basin in between the Cascades and Rocky Mountains because the species requires high elevation meadows near freshwater streams as habitat, which are not found in this region [29]. *M. r. macropus* [24] is found in the northern Rocky Mountains, occupying area in eastern Washington, eastern Oregon, Idaho, Montana, and a small part of northern Utah. *M. r. richardsoni* (sensu DeKay [24]) is found in the Canadian Rocky Mountains. Finally, *M. r. myllodontus* (sensu Rasmussen and Chamberlain [30]) occupies northcentral Utah.

The disjunct distribution of *M. richardsoni* is shared by multiple other taxa in the Pacific Northwest (e.g., [31–34]). Analysis of sequence data from co-distributed vertebrate species such as *Ascaphus* frogs and *Dicamptodon* salamanders suggest that the disjunct distribution in these taxa resulted from an ancient vicariance that occurred before the Pleistocene epoch [22]. However, previous investigations of *M. richardsoni* suggest that the Rocky Mountains populations were founded by a northern Cascades population more recently than the Last Glacial Maximum [22, 34]. Since this work was based on a single mitochondrial gene questions remain, including whether gene flow has occurred across the Columbia Basin, if subspecies boundaries are predictive of population genetic structure, and what demographic changes the species has experienced due to Pleistocene glaciation.

### Samples

Twenty-nine *M. richardsoni* tissue samples were obtained from museums and collaborators (Table 1). Samples represented subspecies *arvicoloides*, *macropus*, and *myllodontus*, and thus represented most of its geographic range (Fig 1). Additional tissue from a single *Microtus montanus* individual was obtained for use as an outgroup. DNA was extracted from all samples using Qiagen DNeasy tissue kits (Qiagen, Hilden, Germany). For analyses requiring *a priori*

**Table 1. Information for all samples used in the present study.** Museum collections: UMNH (Natural History Museum of Utah), DMNS (Denver Museum of Nature and Science), MSB (Museum of Southwestern Biology).

| ID | Source/Catalog # | Subspecies | Region | Lat | Lon | Elevation (m) | Sex |
|---|---|---|---|---|---|---|---|
| ID02 | 1 Sullivan JMG98 | macropus | NRM | 46.33 | -114.63 | | |
| ID03 | 1 Sullivan JMG89 | macropus | NRM | 46.33 | -114.63 | | |
| ID04 | Sullivan JMG243 | macropus | NRM | | | | |
| ID05 | Sullivan JMG244 | macropus | NRM | | | | |
| ID06 | Sullivan JMG245 | macropus | NRM | | | | |
| ID07 | 1 Sullivan JMS275 | macropus | NRM | 45.12 | -116.42 | 1598 | |
| ID08 | 1 Sullivan JMS276 | macropus | NRM | 45.12 | -116.42 | 1598 | |
| ID09 | 1 Sullivan JMS283 | arvicoloides | SC | 45.75 | -121.58 | | |
| ID011 | 1 Sullivan JMS284 | arvicoloides | SC | 45.75 | -121.58 | | |
| ID012 | 1 Sullivan JMS286 | arvicoloides | SC | 44.02 | -121.75 | 1658 | |
| ID013 | 1 Sullivan JMS291 | arvicoloides | SC | 44.02 | -121.75 | 1658 | |
| ID014 | 1 Sullivan JMS292 | arvicoloides | SC | 44.02 | -121.75 | 1658 | |
| ID015 | 1 Sullivan JRD013 | arvicoloides | NC | | | | |
| ID016 | 1 Sullivan JRD064 | macropus | NRM | 46.10 | -117.83 | 1254 | |
| ID017 | 1 Sullivan JRD065 | macropus | NRM | 46.10 | -117.83 | 1254 | |
| ID018 | UMNH Mamm:30848 | myllodontus | UT | 38.20 | -111.58 | 2659 | F |
| ID019 | UMNH Mamm:30891 | myllodontus | UT | 38.20 | -111.58 | 2633 | M |
| ID020 | UMNH Mamm:31124 | myllodontus | UT | 38.44 | -111.46 | 3303 | M |
| ID021 | UMNH Mamm:31148 | myllodontus | UT | 38.44 | -111.58 | 3203 | M |
| ID022 | UMNH Mamm:37701 | myllodontus | UT | 40.69 | -110.90 | 3235 | F |
| ID023 | UMNH Mamm:39998 | myllodontus | UT | 40.68 | -110.92 | 3200 | M |
| ID024 | UMNH Mamm:49000 | myllodontus | UT | 40.64 | -110.95 | 2790 | F |
| ID026 | DMNS Mamm:10074 | macropus | WY | 44.83 | -107.49 | 2633 | M |
| ID027 | DMNS Mamm:10075 | macropus | WY | 44.83 | -107.49 | 2636 | M |
| ID028 | DMNS Mamm:12824 | macropus | WY | 42.93 | -110.53 | 2636 | M |
| ID029 | DMNS Mamm:19027 | macropus | WY | 43.57 | -108.83 | 2700 | F |
| ID030 | A. Chavez ASC531 | arvicoloides | NC | 48.53 | -120.66 | 1653 | F |
| ID031 | A. Chavez ASC532 | arvicoloides | NC | 48.53 | -120.66 | 1653 | M |
| ID032 | MSB: Mamm:143765 | macropus | WY | 43.55 | -111.24 | | F |
| ID033 | MSB: Mamm:227469 | macropus | NRM | 46.68 | -113.62 | 2374 | F |

NRM: Northern Rocky Mountains, SC: South Cascades, NC: North Cascades, UT: Utah, WY: Wyoming, F: Female, M: Male.

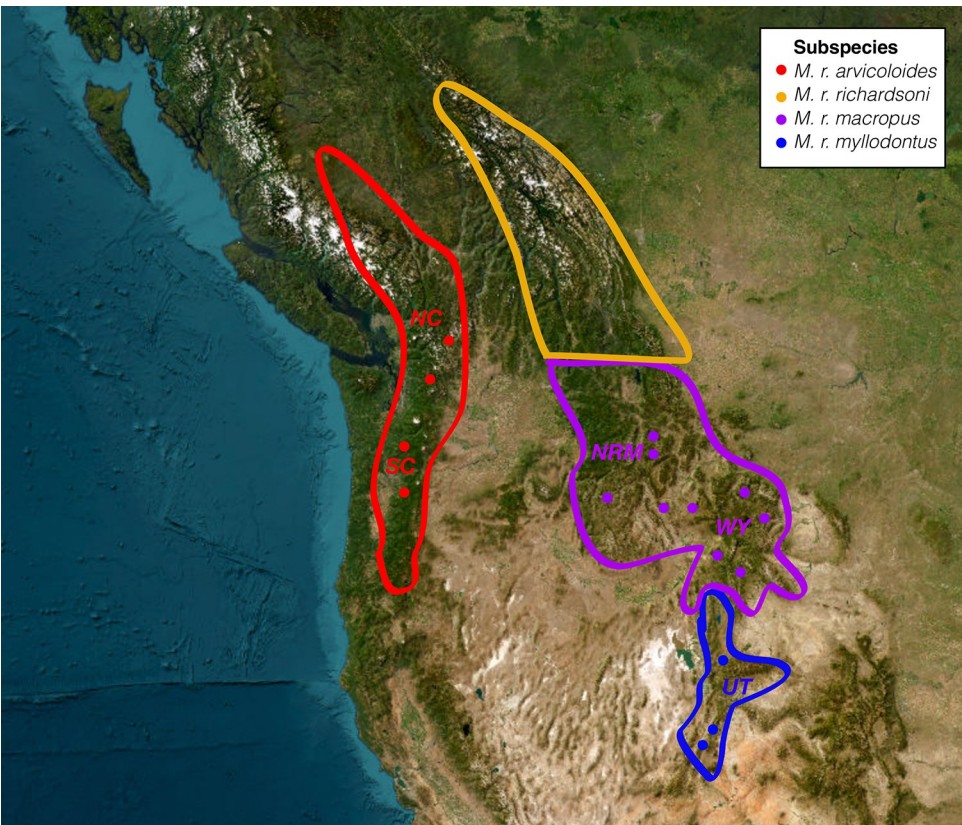

**Fig 1. Range map of *Microtus richardsoni* with sampling locations indicated by black circles.** Subspecies ranges are indicated by different colors and *a priori* groups are labeled. NC: North Cascades, SC: South Cascades, NRM: Northern Rocky Mountains, WY: Wyoming, UT: Utah. Range map recreated from Ludwig, 1984.

population assignment, the samples were grouped into five populations based on subspecies descriptions and results from previous phylogeographic studies with mitochondrial DNA [22, 34]—South Cascades (SC), North Cascades (NC), Northern Rocky Mountains (NRM), Wyoming (WY), and Utah (UT).

## Data collection

**Genotyping by Sequencing (GBS).** Genotyping by sequencing was performed on all samples following a modification of the Elshire et al. [35] protocol using the *PstI* restriction enzyme and performing size selection to obtain fragments between 200 and 500 base pairs in length. Two separate sequencing runs were performed. The first, which occurred during a Research Experiences for Undergraduates program, contained 15 samples, and was sequenced on an Illumina Hi-Seq 2000 at The Ohio State University Comprehensive Cancer Center to produce single-end 50bp reads. The second contained the remaining 15 samples and was sequenced with an Illumina HiSeq4000 at the same facility to produce paired-end 150bp reads. To standardize sequencing reads across the two runs, fastxtrimmer version 0.0.14 was used to remove reverse reads and trim forward reads to 45bp for the second round of sequencing [36]. Variant calling was performed for all samples with iPyrad version 0.9.56 [37] using the *M. r. arvicoloides* genome as a reference (GenBank GCA_020387435.1; [38]. To examine the effect of the reference, variant calling was also performed in iPyrad using de novo clustering with a clustering threshold of 0.85. A principal component analysis showed similar patterns between

the reference and clustering datasets (S1D Fig), but clustering did result in almost twice the number of variants. Separate datasets were generated for both the total dataset (30 samples) and the ingroup only (29 samples). Due to using two different sequencing runs, the number of expected heterozygotes was calculated with VCFtools version 0.1.16 and compared for each sample [39]. A *t*-test performed on the expected number of homozygous sites per sample using the t.test function in R version 3.3.1 [40] showed no significant difference between the two sequencing runs ($t$ = -1.7244, df = 24.11, $p$ = 0.09743). Additionally, variant SNPs were called with a subsampled dataset. Samples from the higher coverage sequencing run were subsampled to the average number of reads of the lower coverage sequencing run using seqtk version 1.3-r115-dirty (https://github.com/lh3/seqtk), variant calling was performed as below, and PCA showed very little difference between the full and subsampled datasets (S1B Fig). Therefore, the full, reference-guided dataset was used for all subsequent analyses.

Variant calling in iPyrad used a minimum depth per sample per locus of 8, a minimum locus length of 35bp, and a minimum number of samples per locus of 15 (~50%). The resulting VCF file was filtered using VCFTools, with a minimum number of samples per SNP of 22 (~75%). The mean depth per SNP was calculated with VCFTools and plotted with the Barplot function in R. A long tail of high mean depths suggested possible poor read mapping due to genomic repeats. To filter out SNPs with these high values, the maximum mean depth per SNP was set to 107. This value was calculated by the equation:

$mean\ of\ mean\ depths\ per\ SNP + 3\sqrt{mean}$, as suggested in the ddocent documentation [41]. Individuals missing greater than 30% of sites were removed, and only biallelic SNPs were retained. For analyses that may be significantly affected by linkage among SNPs, two methods were used. First, a dataset was generated in which a single SNP was randomly selected from each locus using a custom python script (https://github.com/djlduckett/Genome_Resources/blob/master/1snp_by_id.py). For the second dataset, Plink version 1.90b6.12 was used to calculate correlations among SNPs using stepwise sliding 50 variant windows with a step size of 5 variants, and one SNP was randomly chosen when the correlation coefficient ($r^2$) between a pair of SNPs was greater than a predefined threshold [42]. Thresholds of 0.3 and 0.6 resulted in very similar principal component results (S1C Fig), so 0.6 was used for subsequent analyses to retain more SNPs in the dataset.

**Whole genome sequencing.** Whole genome sequencing was conducted for two samples and the outgroup in a previous study [38]. Whole genome library preparation and sequencing was performed on the remaining 27 samples with 150bp paired end reads on an Illumina NovaSeq by the University of California Davis Genome Center. Variant calling was performed with GATK for both total and ingroup-only datasets. Raw reads were trimmed with Trimmomatic version 0.36 with settings ILLUMINACLIP: 2:30:10, LEADING:3, TRAILING:3, SLIDINGWINDOW:4:15, and MINLEN:36 [43]. Trimmed reads were mapped to the *M. r. arvicoloides* reference genome with bwa version 0.7.17-r1198 [44]. bwa was used to add read groups and map trimmed reads to the *M. r. arvicoloides* reference genome. Picard version 2.3.0 (http://broadinstitute.github.io/picard) was used to remove PCR duplicates.

Variants for each sample were called using HaplotypeCaller and variants among all samples were called using GenotypeGVCFs. SNPs were separated from INDELs using GATK's separate_variants function, and the resulting SNPs were filtered using GATK best practices, removing SNPs with the following settings: depth normalized variant confidence (QD) < 2.0, mapping quality (MQ) < 40, strand bias estimate (FS) > 60.0, base quality (QUAL) < 30, median mapping quality for each allele (MQRankSum) < -12.5, position bias (ReadPosRankSum) < -8.0, and strand bias (SOR) > 3.0 [45]. Additionally, VCFTools was used to remove SNPs present in less than 22 samples (75%) and samples were removed if they had greater than

30% missing data. Non-biallelic SNPs were removed and Plink was used to filter SNPs for linkage disequilibrium as with the iPyrad dataset. Similar to the iPyrad dataset, variant calling and filtering was performed separately for total and ingroup-only datasets.

**Ultraconserved elements (UCEs).** UCEs were extracted from the trimmed whole genome sequencing reads. For each sample, UCE assemblies were performed with *itero* version 1.1.2 using the trimmed reads as input and loci from the *uce-5k-probes* tetrapod dataset as a reference (https://github.com/faircloth-lab/itero; http://www.ultraconserved.org). To construct a reference UCE dataset to use for variant calling, assembled contigs were matched to UCE probes and UCE loci were extracted using Phyluce version 1.7.1 [46]. Next, a custom Python script was used to create the reference by extracting the longest sequence for each locus that contained less than 10% missing data (https://github.com/djlduckett/Genome_Resources/blob/master/build_uce_reference.py). To better emulate typical UCE locus sequencing, sequences longer than 1000bp were excluded. After completing the reference, variant calling was performed with GATK as above. One SNP was randomly sampled per UCE locus using the same script as for subsampling the GBS iPyrad SNPs.

**Mitochondrial markers.** Mitochondrial genomes for each of the samples were assembled by mapping the low coverage genome reads to the *M. r. arvicoloides* reference mitochondrial genome (GenBank MW387162; [38]). All mitochondrial genomes were aligned with Muscle in Geneious v. 9.1.8 [47, 48], and the COI and cytb mitochondrial genes were extracted based on coordinates from the *M. r. arvicoloides* reference annotation using Geneious. Species identification was checked for all samples using the COI gene and the BOLD database [49]. The number of total sites and number of polymorphic sites were calculated in Geneious.

## Phylogeographic analyses

**Population structure.** Autosomal expected heterozygosity was calculated for each nuclear data type. Expected heterozygosity per site was calculated from each VCF file using VCFtools, the heterozygosity was summed across sites, and the sum was divided by the total number of sites (invariant and variant). To examine population structure, principal component analysis (PCA) was performed on each nuclear ingroup dataset using the SNPRelate package version 1.8.0 in R [50]. Analyses of Molecular Variance (AMOVAs; [51]) were also performed in R by converting VCF files to STRUCTURE format with PGDSpider version 2.1.0.3 ([52]), reading STRUCTURE files with the adegenet package version 2.1.0 [53], and conducting AMOVAs with the poppr package version 2.6.0 [54]. Significance of the AMOVAs was assessed using 1000 permutations with the randtest function [55]. AMOVAs were performed in R for both cytb and whole mitochondrial datasets as above but reading the alignments into R with the apex package version 1.0.4 (https://github.com/thibautjombart/apex). Due to its large size, the AMOVA using the whole genome dataset did not finish despite being run on a computing cluster with 28 cores and 112GB RAM for longer than two weeks. To overcome this limitation, we randomly sampled 100,000 SNPs from the STRUCTURE file using a custom Python script (https://github.com/djlduckett/Genome_Resources/blob/master/subsample_structure.py) and analyzed this subsampled dataset. While this subsampling artificially decreases the size of the WGS dataset, the randomly sampled data still provide a good comparison to the GBS datasets, where the main theoretical concern is whether the restriction digested data adequately represent the whole genome. In addition to an AMOVA, population structure was analyzed for the nuclear datasets using the maximum likelihood clustering approach implemented in ADMIXTURE version 1.3.0 [56]. Ten repetitions per number of clusters were performed using ADMIXTURE varying the number of clusters from one to ten, and the number of clusters with the lowest cross validation error was chosen as best. PONG version 1.4.9 was used to visualize ADMIXTURE results [57].

**Historical demography.** Stairway plots (version 2.1.1; [58]) were conducted to examine demographic history over time for the nuclear datasets. For each nuclear ingroup dataset, the site frequency spectrum was generated with easySFS (https://github.com/isaacovercast/easySFS). Although population structure and other evolutionary factors can have a significant effect on the interpretation of these types of analyses (e.g., [59–62]), we consider the stairway plots as genomic summaries of evolutionary processes and follow recent suggestions to not interpret the results as analogous to census population sizes (e.g., [63, 64]). Using this interpretation provides a useful way to compare the results that can be obtained using different molecular markers for the same system. Therefore, we included all samples when producing site frequency spectra. Due to missing data, the preview function in easySFS was used to down project the site frequency spectra, retaining the largest number of SNPs. Stairway plots were plotted using a custom R script provided by E. Fonseca (*pers. Comm.*; included in supplemental materials) with the generation time set to one year [22, 29]. The mutation rate was set to $1.1\times10^{-8}$ substitutions per site per generation for the nuclear datasets [60, 65].

For mitochondrial datasets, best models of nucleotide substitution were assessed using autoModel in PAUP* version 4.0 with corrected Akaike Information Criterion [66]. Bayesian Skyline analyses were performed with BEAST version 2.5.0 [67], with a generation time of one year and a mutation rate of $4.572\times10^{-7}$ substitutions per site per generation [68]. The Markov chain was run for 10M steps, and convergence was assessed using trace plots and effective sample size (ESS) estimates in Tracer version 1.7.1 [69]. Results were plotted with the same R script as above for plotting Stairway plots. Because mitochondrial Bayesian Skyline analyses rely on a single gene tree, a gene tree was estimated from the mtgenome data using maximum likelihood as implemented in PAUP* with the model of nucleotide substitution chosen above to use as input.

**Genetic distance.** To obtain genetic distances among samples for each data type, SNPs from the total nuclear datasets were concatenated and converted to nexus format with vcf2phylip (https://github.com/edgardomortiz/vcf2phylip). All genetic distances were calculated using PAUP*. Nuclear data types used uncorrected *p*-distances and mitochondrial data types used likelihood corrected distances using the models of sequence evolution assessed above. To examine differences among marker types, heatmaps were constructed with the ggplot2 R package version 3.3.0 [70]), with the top half of the matrix referring to distances among individuals within a data type and the bottom half referring to the differences in distances between the selected data type and the WGS distances. Additionally, pairwise Mantel tests were performed among all datasets to identify correlations among the distance matrices [71]. These tests were performed using the vegan package version 2.5–7 in R, and significance was assessed using 999 permutations [72].

## Cost comparison

A cost comparison was performed among all marker types. GBS and WGS costs were obtained from empirical data collected above. Mitochondrial cytb costs were based on previous cytb sequencing performed in other taxa. Whole mitochondrial genome costs were calculated as if they were obtained using a whole mitochondrial genome hybridization capture approach (e.g., [73]). Both mitochondrial genome and UCE cost estimates were obtained from labs experienced with these approaches (Brant Faircloth and Lei Yang, *pers. comm.*). Importantly, UCE cost estimates did not include probe development and thus assume that probes exist for the focal taxon. Comparisons were made with the total datasets (30 samples), and included the total number of sites per sample, number of SNPs, total price for all samples, price per sample, price per SNP, and price per sample per SNP. Computational costs were also calculated for

applicable analyses, (i.e., those conducted as batch processes using a computing cluster). These consisted of data preparation, ADMIXTURE analyses, and Skyline/Stairway plots. All computation was performed using a single node with 28 cores and 128 GB RAM at the Ohio Supercomputer Center. Compiled values included the number of CPU hours and the number of walltime hours. To mirror the process of obtaining only whole mitochondrial genomes when calculating computational cost, mitochondrial reads were extracted that aligned to the reference mitochondrial genome with mapping quality > 30 using SAMTOOLS version 1.10, and the resulting reads were assembled with SPADES version 3.15.3 [74]. The same process was also performed for the UCE data to better estimate the computational cost as if UCE reads were sequenced directly rather than extracting them from WGS data.

## Results

### Data collection

GBS variant calling with iPyrad produced ingroup datasets with 30,087 SNPs when sampling one SNP per locus and 27,679 SNPs when filtering for linkage disequilibrium. The datasets had an average depth per individual of 74.14 and 74.39 (range 13.79–171.43 and 13.78–192.65), and average missing data per individual of 1.40% and 1.49% (range 0.11–9.72 and 0.11–10.20) for the one SNP per locus and linkage filtered datasets respectively. Subsampling reads to obtain even coverage among samples did not affect PCA results (S1B Fig). Without filtering for linkage disequilibrium, the GBS-iPyrad and WGS datasets contained 40,608 and 31,558,250 SNPs respectively. GBS-iPyrad shared 37,785 SNPs with the WGS dataset, with the GBS-iPyrad dataset having 2,823 exclusive SNPs. Whole genome sequencing produced an average raw coverage of 12.7x (range 10.1–15.9). Variant calling and filtering resulted in 10,379,876 SNPs from 29 ingroup samples, with an average depth per individual of 12.13 (range 8.18–31.81), 0.38% missing data per individual (range 0.03–0.80), and an expected heterozygosity of 0.0030.

Assembling UCEs from genome sequencing data resulted in an average of 3827.2 loci (range 3299–4494), with a median length of 360.8 bp (range 308–1783). The reference UCE set consisted of 4125 loci with mean length 394 bp (range 199–987). Variant calling from the UCE loci produced 3735 SNPs, with an average depth per individual of 14.14 (range 9.23–37.22), average missing data per individual of 1.84% (range 0.40–13.07) and expected autosomal heterozygosity of 0.00022.

Mitochondrial sequences were successfully extracted from the whole genome sequencing reads for all samples, with sequence lengths of 1143bp and 16,285bp for cytb and whole mitochondrial genomes respectively. Correct species identities were confirmed for all samples when analyzed with BOLD.

### Phylogeographic analyses

**Population structure.** PCA results derived from all the reduced representation datasets were similar (Fig 2). All analyses showed separate clusters for the South Cascades, North Cascades, Northern Rocky Mountains, Wyoming, and Utah as well as a distinct separation of the Big Horn Mountains from the rest of the Wyoming samples. There were a few minor differences, for example, the UCE results did not clearly separate the Utah samples from the Wyoming samples. PCA analysis from the whole genome dataset was similar to those seen with the GBS iPyrad and UCE datasets (Fig 2D). All datasets displayed a small number of intermediate samples. Samples from the Blue Mountains (Washington) were found between the Cascades groups and the Northern Rocky Mountains, and a sample from Lemhi, Idaho, was found between the Northern Rocky Mountains and Wyoming.

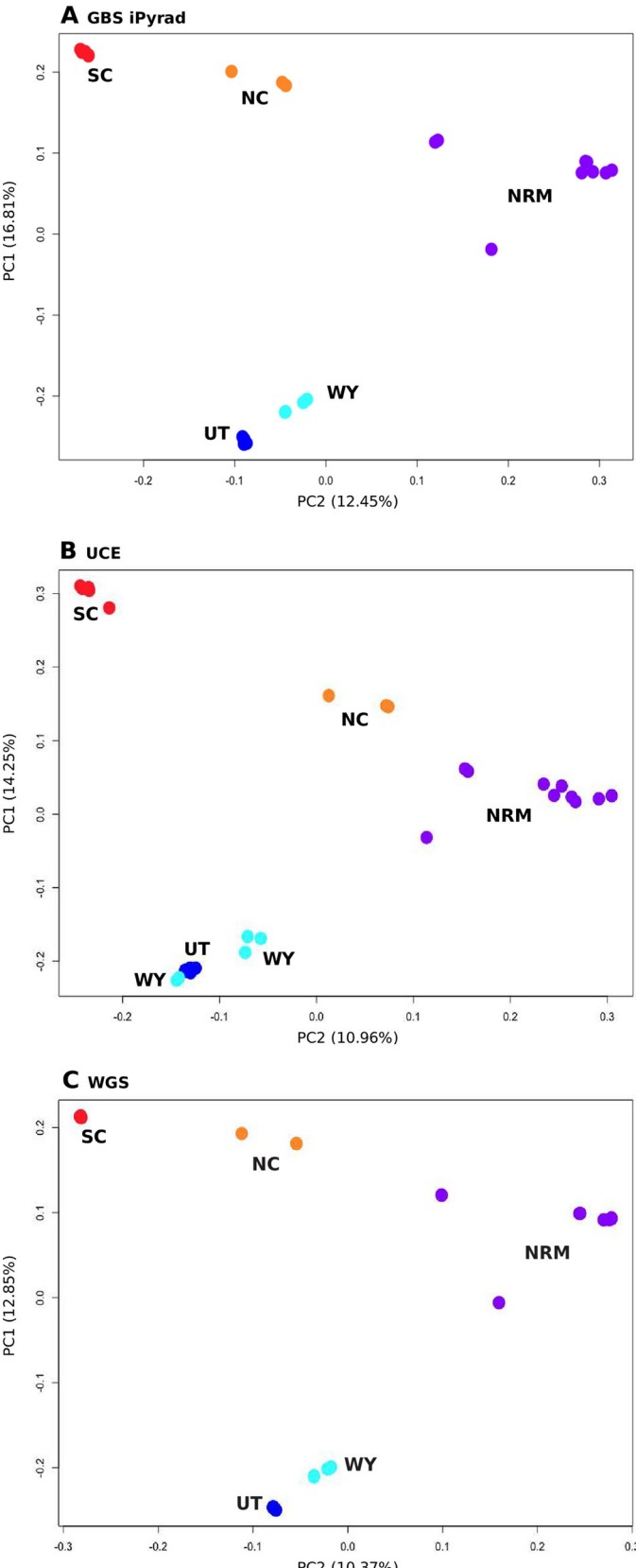

**Fig 2. PCA results from nuclear datasets, with *a priori* groups labeled and in different colors.** SC: South Cascades, NC: North Cascades, NRM: Northern Rocky Mountains, WY: Wyoming, UT: Utah. Datasets include A) GBS iPyrad, B) UCE, and C) WGS.

AMOVA results from both nuclear and mitochondrial datasets indicate that most of the genetic variation can be explained by individual differences within populations (S1 Table). Both within and among populations distinctions explained a significant amount of the genetic variance for the analyzed data except for that of the whole mitochondrial genomes, where among population variance was not significant. In AMOVAs where samples were divided by subspecies this division did not explain a significant amount of genetic variance for any dataset. With ten repetitions each, ADMIXTURE analyses were again similar between the GBS iPyrad and UCE datasets, but not identical (Fig 3, S3 Fig). The GBS iPyrad dataset identified the optimal number of clusters (K) to be four. With four clusters, all replicates contained Cascades, Rocky Mountains, and Utah clusters. However, five replicates included Wyoming as a separate group, four replicates included the Blue Mountains as a separate group, and one replicate included the North Cascades as a separate group. The remaining nuclear datasets were consistent in finding K = 3. These clusters consisted of Cascades and Rocky Mountains clusters, and a cluster containing both Wyoming and Utah. Similar clustering was obtained when examining the GBS iPyrad dataset with K = 3. When examining K = 4 in the nuclear datasets, the identity of the most supported fourth cluster varied by dataset, with GBS iPyrad and WGS supporting a separate Wyoming cluster, and the UCE datasets supporting a separate North Cascades cluster.

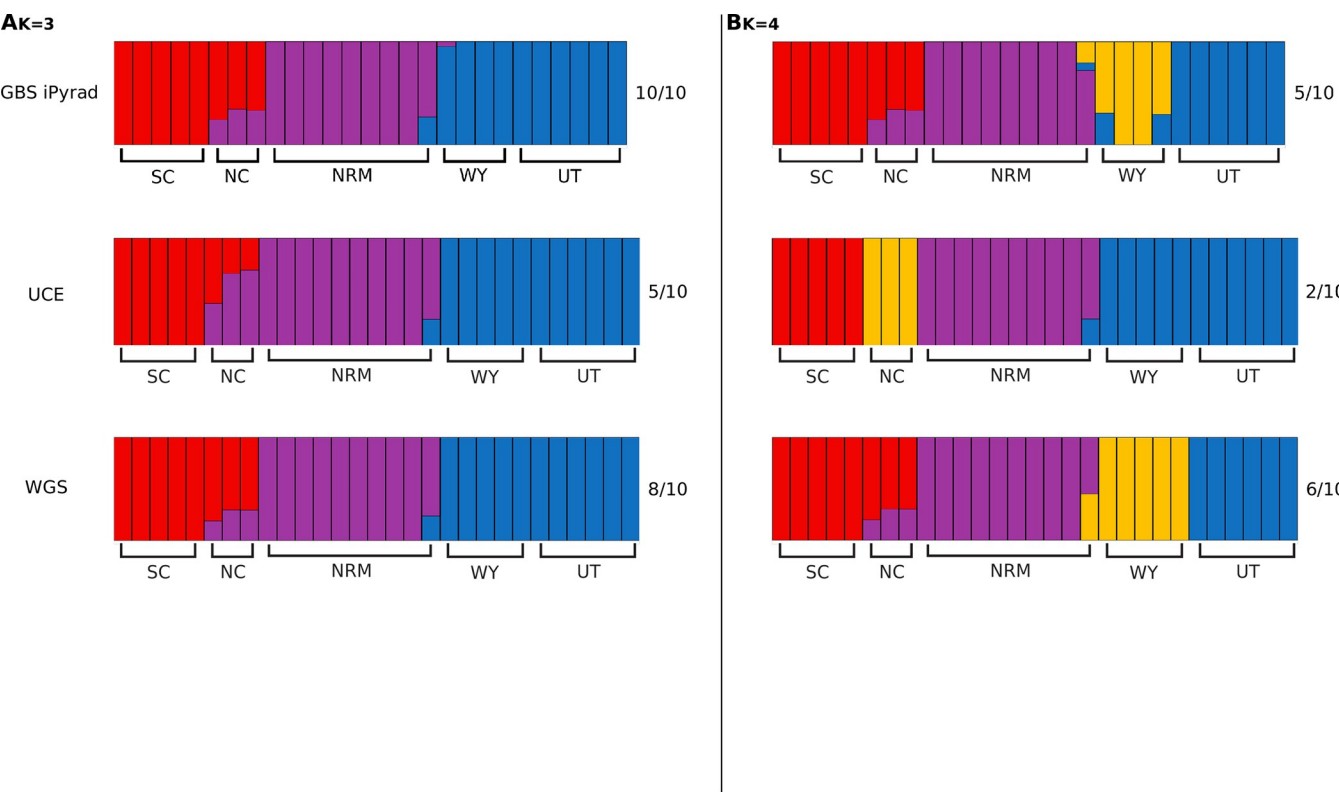

**Fig 3. ADMIXTURE results for nuclear datasets.** Three and four cluster plots are shown for each dataset. The optimal number of clusters was four for the GBS iPyrad dataset and three for the UCE and WGS datasets. Number of repetitions out of ten supporting each plot is shown below the value of K. A) GBS iPyrad, B) UCE, and C) WGS. SC: South Cascades, NC: North Cascades, NRM: Northern Rocky Mountains, WY: Wyoming, UT: Utah.

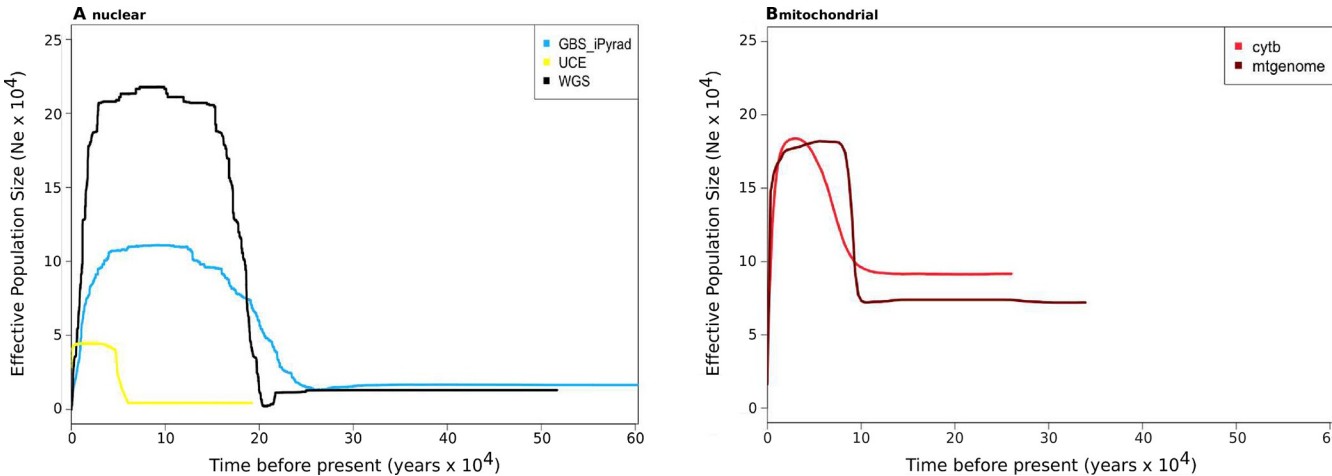

**Fig 4. Results for historical demography analyses.** Lines represent the estimated median value of Ne. Mitochondrial datasets were analyzed using Bayesian Skyline Plots and nuclear datasets were analyzed using Stairway Plots. Generation time = 1 year, mitochondrial μ = 4.572x10^{-7} subs/site/gen and nuclear μ = 1x10^{-8} subs/site/gen. Mitochondrial Ne values were scaled by four to account for differences in marker effective population size. Plots with confidence intervals can be found in S4 Fig.

**Historical demography.** All nuclear datasets exhibited a similar pattern of changes in effective population size (Ne) through time (Fig 4). Apart from the UCE dataset (below), results from each showed an increase in Ne around 20k-30k years ago, a plateau at ~10k years ago, and then a sharp decline towards the present (S4 Fig). The UCE dataset displays a similar overall pattern, but the estimated Ne values are smaller and the time of increase is much closer to the present than the other nuclear datasets. With the mitochondrial markers, the best substitution models as assessed through PAUP* were TrN+I and GTR+I+G for cytb and the mtgenome, respectively. The mtgenome likelihood phylogeny included the South Cascades as a monophyletic clade divergent from all other samples, and the North Cascades samples were included in a clade with the Northern Rocky Mountains samples (S5 Fig). Skyline plots from the cytb and mtgenome datasets were very similar to each other. After scaling for differences in effective size between nuclear and mitochondrial markers, the Skyline plot inferred from the mtgenome data exhibits a similar pattern to the GBS and WGS datasets, albeit with a more recent inferred timing of the plateau, starting ~10k years ago.

**Genetic distance.** Overall, the largest genetic distances among individuals are generally between samples from the South Cascades and Utah (Fig 5). Distances from different mitochondrial datasets are similar, with little distance seen among individuals, but large distances between ingroup individuals and the outgroup. The mitochondrial data produced lower estimates of genetic distance among individuals than did the WGS data, with maximum ingroup distances of 0.047, 0.038, 0.191, 0.077, and 0.156 for cytb, mtgenome, GBS iPyrad, UCE, and WGS datasets respectively. WGS and GBS iPyrad data resulted in higher estimates of the genetic distance among individuals, and the estimates of genetic distance from GBS were more similar to those from WGS than the mitochondria. The UCE data type exhibits similar patterns to both the mitochondrial and GBS datasets, as distances among individuals in the UCE dataset and between the UCE and WGS datasets are low, while the UCE dataset shows the highest distance between outgroup and ingroup samples of any data type. Mantel tests reflected similar patterns. All distance matrices were significantly correlated (α = 0.001; S2 Table). As shown by $r^2$ values, cytb distances were more predictive of mtgenome distances (0.9968) than nuclear distances, and GBS distances were more predictive of WGS distances

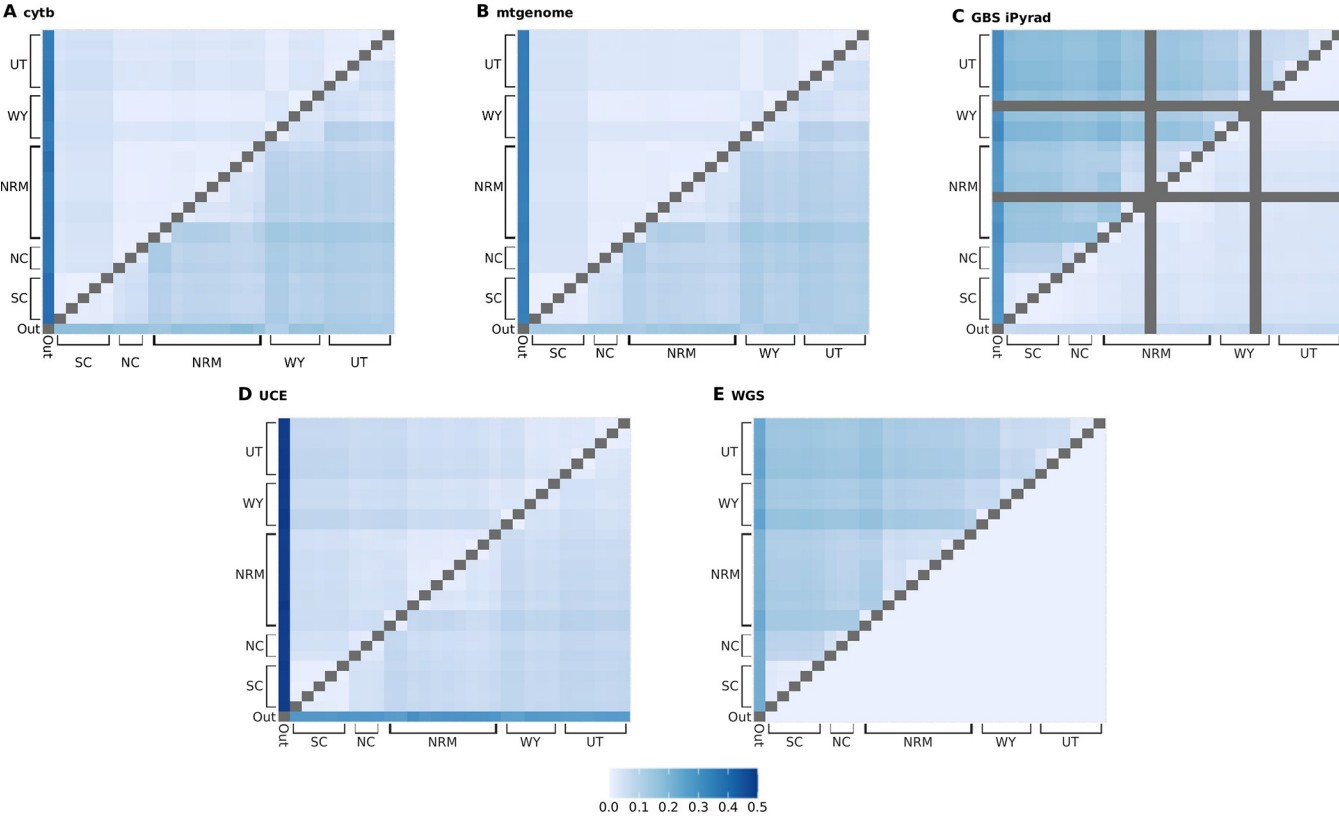

**Fig 5. Genetic distances among samples.** For each heatmap, the top triangle represents the distances among samples within a data type, and the bottom triangle represents the difference in distances between the current data type and the WGS distances. A) cytb, B) mtgenome, C) GBS iPyrad, D) UCE, E) WGS. SC: South Cascades, NC: North Cascades, NRM: Northern Rocky Mountains, WY: Wyoming, UT: Utah. Note: the GBS iPyrad dataset has two fewer samples than the other data types, and distances including these samples are grey in heatmap C.

than any other marker (0.9912 for GBS iPyrad). UCE distances were more predictive of mtgenome distances than WGS distances (0.9860 compared to 0.5654 respectively).

## Cost comparison

Whole genome sequencing was the most expensive and mitochondrial cytb sequencing was the least expensive overall (Table 2) in terms of the total cost to obtain the data. Obtaining whole nuclear genomes was about 16 times as expensive as performing GBS while obtaining UCEs was over twice as expensive as GBS. However, when calculated on a per SNP basis these relationships are inverted, with WGS being substantially more cost effective on a per SNP

**Table 2. Cost comparison among different data types.** Number of SNPs refers to the ingroup dataset (29 samples) after filtering, but not filtering for linkage.

| Marker | Mb/sample | SNPs | Total Price (30 samples) | Price/ sample | Price/ SNP | Price/sample/ SNP | CPU Time (hrs) |
|---|---|---|---|---|---|---|---|
| cytb | 1.14 | 72 | $124 | $4.25 | $1.72 | $0.06 | 9.96 |
| mtgenome | 16.29 | 1101 | $1650 | $55.00 | $1.50 | $0.05 | 74.50 |
| GBS iPyrad | 6,777.75 | 40,608 | $725 | $25.00 | $0.02 | $0.0006 | 95.29 |
| UCE | 1,472.22 | 19,334 | $2010 | $67.00 | $0.10 | $0.003 | 2608.09 |
| WGS | 2,140,065.55 | 31,558,250 | $11,660 | $412.00 | $0.0004 | $0.00001 | 44856.66 |

basis than GBS or UCE data. Computational cost generally increased with an increasing number of loci among data types (S3 Table), with the exception of UCEs having fewer loci and higher computational costs than the other reduced representation data type. Notably, most analyses reported here could have been performed using a desktop computer rather than a computing cluster, except for analyses of the whole nuclear genome data.

## Discussion

In this investigation we conducted some common phylogeographic analyses using different mitochondrial, reduced representation nuclear, and whole genome nuclear datasets. Our goal was to understand how results from an empirical system differ as a function of marker type and to conduct a cost-benefit analysis that could guide research study design. Since we are primarily interested in how different markers may (or may not) lead to different inferences about the demographic history of the focal taxon, we compare and contrast these results in the context of *M. richardsoni*'s demographic history.

### Genetic distances among samples

Genetic distances among samples estimated from different nuclear datasets were broadly similar. For example, the distances estimated from the GBS data were almost as predictive of WGS distances ($r^2$ = 0.9912; $r^2$ = 0.7955) as the cytb distances to the mitochondrial genome data ($r^2$ = 0.9968). This result offers confirmation that GBS and other reduced representation datasets are effective at randomly sampling the nuclear genome. However, the distances estimated from the UCE data were not similar to those inferred from other nuclear data ($r^2$ = 0.5654), as among individual distances were lower and distances between the outgroup and ingroup samples were higher than those estimate from other nuclear data. With mutation rates up to 20 times less than general nuclear rates [75, 76], the slower rate of UCE evolution combined with the recent divergence among populations within *M. richardsoni* [22, 34], likely caused the small genetic distances among ingroup samples. While this result might imply that UCE data are less useful for phylogeographic investigations then SNP data (although see [77, 78]), it supports the assertion that these data are effective for phylogenetic analysis [79].

### Population genetic structure

Results from all data sets demonstrate that there is population genetic structure within *M. richardsoni*. This is consistent with results from previous work in the system and reflects the natural history of *M. richardsoni*, specifically their small body size and fidelity to high-elevation freshwater streams [29]. However, only some of the population genetic structure identified using different markers reflects the described subspecies. For example, we recover genetic structure that corresponds to *M. r. arvicoloides* in the Cascades Mountains and *M. r. macropus* in the Rocky Mountains, but our results are not congruent with other subspecies descriptions. We also did not find that genetic variance was partitioned across subspecies in the AMOVA analysis.

Results from different nuclear markers from both the PCA and the ADMIXTURE analyses indicate that Wyoming samples cluster with Utah individuals and suggests that Wyoming individuals are more closely related to *M. r. myllodontus* than *M. r. macropus*. The subspecies *M. r. myllodontus* is known only from Utah and extreme southern Idaho [27, 80], but these results support the hypothesis that the two subspecies may overlap in Wyoming, as posited in the original subspecies description [30]. In mammals, subspecies classifications often represent geographic clusters based on labile morphological characters [81], and therefore may be poor predictors of phylogeographic structure. Further analysis with additional field sampling is

necessary to determine whether the observed pattern is the result of previously unobserved *M. r. myllodontus* individuals in Wyoming, phylogenetic relatedness between the Wyoming and Utah individuals, or gene flow between the two groups.

## Historical demography

There is broad similarity across marker types in the analysis of historical demography as measured by effective population size change (Ne). Nearly all data types exhibit a plateau in effective population size within the last ~20k years. This change in Ne is likely caused by changes in both effective population size and in population structure and gene flow [61] that coincide broadly to the Last Glacial Maximum (LGM). Much of the current species range was glaciated at the time of the LGM, and this event has been previously implicated in having a substantial impact on *M. richardsoni*'s demographic history [22, 34]). While the broad trends are similar, there were subtle differences across results from different marker types. For example, results from the GBS dataset showed the initial increase in Ne slightly earlier than the whole genome dataset, and each dataset displayed a slightly different peak in Ne, with the plateaus reaching ~110k and ~220k for the GBS iPyrad and WGS datasets, respectively. However, for these analyses the 95% confidence intervals are large enough to encapsulate values for each of the nuclear datasets (S4 Fig). Moreover, one subspecies, *M. r. richardsoni*, is missing from the present study due to sampling constraints, an omission which potentially could influence the demographic modeling conducted here. While comparisons among markers should be robust to such sampling constraints, the inclusion of this subspecies could affect interpretation of demographic history for the species.

Inferences about demographic size change were similar across molecular markers. For example, Stairway plots from GBS and WGS datasets showed similar patterns, with broad overlap in confidence intervals and only slight differences in the point estimates of the magnitude and timing of demographic change. Our results contrast with previous comparisons of RADSeq and whole genome sequencing with pairwise sequentially Markovian coalescent (PSMC) analyses [82, 83], likely because the stairway plots use the site frequency spectrum as a summary of SNP information rather than linkage among sites. UCE stairway plots differed greatly from the WGS and GBS datasets in the magnitude and timing of changes in Ne, potentially due to a lack of understanding about how to model mutation rate in ultraconserved elements.

Mitochondrial skyline plot analyses display a similar demographic pattern to nuclear stairway plot analyses, although the timing of events was much more recent in the mitochondrial data. The difference in marker properties, analytical programs used for analysis, and assumed mutation rates with inherent error rates likely prevent these demographic results from being directly comparable, but the similar patterns in the magnitude of change may indicate that a strong signal is being detected by both marker types. Researchers would be likely to make the same general inference about the historical demography of *M. richardsoni* from either of these analyses regardless of the data that they were based upon.

## Mitochondrial genomes

Modern sequencing technologies make is feasible to sequence mitochondrial genomes as a byproduct of WGS or sequence capture protocols. Since the mitochondrial genome does not recombine, researchers may question whether it is worthwhile to target mitochondrial genomes during their sequencing experiments, particularly if single locus mitochondrial data are available. We found that many results were similar when comparing results from a single mitochondrial gene (cytb) and the complete mitochondrial genome. For example, the

AMOVA results from both datasets indicate that most of the genetic variation was explained by within population variance, although population structure among populations was statistically significant when the cytb only dataset was analyzed but not when the analyzing the complete mitochondrial genome. This difference in resolution is likely an artefact of sample size, as the latter necessarily has more variant sites than the former. Genetic distance matrices among samples were broadly similar in both data sets (Fig 5). Given that obtaining mitochondrial genomes is expensive compared to the costs associated with sequencing a single mitochondrial genes, mitochondrial genomes may not justify a special investment. However, if the mitochondrial genome can be assembled from existing WGS reads then the assembly of these data is likely to be worth the extra computational effort, particularly because complete mitochondrial genomes are informative for other analyses, for example distance networks and phylogenetic trees [84–86].

## Reduced representation vs WGS approaches

One of the most promising findings of this work is that only small differences were evident in results from the reduced representation (GBS iPyrad, UCE) and whole genome datasets. All nuclear datasets (GBS iPyrad, UCE, and WGS) produced similar results with PCA and ADMIXTURE, generally agreeing that there were at least three distinct populations; one in the Cascades Mountains, one in the Northern Rocky Mountains, and one in Wyoming and Utah. The optimal number of ADMIXTURE clusters was three for all nuclear datasets except GBS iPyrad, but clustering of samples with K = 3 was very similar across all nuclear datasets. Additionally, clustering was similar across datasets with K = 4, except for which sampling locations constituted the fourth cluster, with GBS iPyrad and WGS separating a Wyoming population and the UCE dataset separating North Cascades individuals. This represents a slight difference in resolution rather than a different interpretation of the population genetic structure. PCA analysis for each marker type generally clustered individuals by sampling location, but there was more resolution when SNPs from GBS or WGS were used, as data collected from the UCE markers did not partition Wyoming and Utah. We suspect that this reflects the slower mutation rate in the UCE loci because our historical demographic inferences indicate that the Utah and Wyoming clusters are the most recently diverged. Nevertheless, the similarity in population structure results across marker types suggests that SNPs from any source are useful in inferring population genetic structure.

## Financial and computational cost comparison

The costs associated with collecting whole genome sequences far exceed those for reduced representation data. Given the similar results among nuclear data types, reduced representation sequencing represents a cost-effective (i.e., financial and computational) method for research investigations that utilize any of the phylogeographic analyses presented here. Our results are consistent with previous comparisons between WGS and RAD Capture (Rapture; [87]) and population structure and phylogeny [88]. However, other evaluations compared analyses for which linkage among sites is considered, including runs of homozygosity [20], divergence landscapes [21], and some aspects of demographic modeling [19], and indicated that WGS data offered better results than SNPs from reduced representation sequencing. As such it is likely that the lack of apparent advantage with WGS noted in our study is due to the fact that the analyses presented here do not incorporate linkage information. Additionally, the lower cost of reduced representation methods allows researchers to increase the number of individuals sampled in comparison to WGS, which is likely to provide increased power and

resolution. In this study, for example, SNP data from over 400 samples could have been collected using GBS for the same price as whole genome sequencing on 30 samples.

Computational effort represents another axis of consideration. UCE processing requires slightly more computational effort than the GBS pipelines, mostly because the raw reads must be assembled *de novo* with Spades before being matched to UCE probes. However, the whole genome data required substantially more computational effort. Not only did analyzing the WGS data use ~364x more core hours than GBS, but it also required multiple terabytes of storage space, and both core hours and storage are expensive and potentially difficult for researchers to access. Depth of coverage has a large impact on both sequencing cost and analysis. The genome sequencing presented here represents intermediate coverage (~12x). Lowering this coverage would allow a researcher to add more samples for the same cost, which some studies have indicated to be advantageous [20, 89]. However, low coverage sequencing requires analyses to incorporate genotype likelihoods or population allele frequencies rather than relying on individual genotypes. While this input is becoming increasingly common, it is still not supported for some analyses, including those assessing demography [89]. Genome sequencing does have an advantage in that it produces reads from the mitochondrial genome as a byproduct that can be used to assemble the mitochondrial genome at little extra cost [90].

## Conclusions

We used five different datasets to compare results from common phylogeographic methods for estimating genetic distance, assessing population structure, and inferring historical demography. We find population structure and patterns of historical demography were similar among most marker types, but genetic distance results differed substantially between mitochondrial and nuclear data. While whole genome sequencing produces substantially more data than reduced representation methods, it is likely only cost-effective if the hypotheses being investigated is better tested via the results of analyses that use linkage information and/ or natural selection. For "*traditional*" phylogeographic questions that utilize AMOVAs, PCAs, ADMIXTURE analyses, Stairway plots, and approaches using genetic distances, our results indicate that reduced representation methods produce similar results to whole genome sequencing at a fraction of the cost.

## Supporting information

**S1 Table. AMOVA results.**
(PDF)

**S2 Table. Pairwise Mantel test results.**
(PDF)

**S3 Table. Computational cost associated with each data type.**
(PDF)

**S1 Fig. PCA Analyses with different ipyrad settings.**
(PDF)

**S2 Fig. Venn diagrams displaying the number of shared versus exclusive SNPs.**
(PDF)

**S3 Fig. Full ADMIXTURE results for each nuclear data type for K = 3 and K = 4.**
(PDF)

**S4 Fig. Individual Skyline and Stairway plots for each marker type, with confidence intervals.**
(PDF)

**S5 Fig. Maximum likelihood mtgenome phylogeny.**
(PDF)

## Acknowledgments

We thank J. Good, J. Demboski at the Denver Natural History Museum, E. Rickart at the Utah Museum of Natural History, J. Cook and J. Dunnum at the Museum of Southwestern Biology, and A. Chavez for tissue samples. We thank B. Faircloth UCE cost estimates and UCE assembly suggestions, L. Barrow for discussions about UCEs, L. Yang for mitochondrial genome cost estimates, M. Smith for discussions about collecting and analyzing data, E. Fonseca for sharing an R script for plotting stairway results, members of the Carstens Lab and R. Norris for manuscript suggestions, and two anonymous reviewers for manuscript feedback.

## Author Contributions

**Conceptualization:** Drew J. Duckett, Jack Sullivan, David C. Tank, Bryan C. Carstens.

**Data curation:** Drew J. Duckett, Kailee Calder.

**Formal analysis:** Drew J. Duckett, Jack Sullivan, David C. Tank.

**Funding acquisition:** Jack Sullivan, David C. Tank, Bryan C. Carstens.

**Investigation:** Drew J. Duckett, Kailee Calder, Bryan C. Carstens.

**Methodology:** Bryan C. Carstens.

**Project administration:** Bryan C. Carstens.

**Software:** Drew J. Duckett.

**Supervision:** Bryan C. Carstens.

**Validation:** Drew J. Duckett.

**Visualization:** Drew J. Duckett.

**Writing – original draft:** Drew J. Duckett, Bryan C. Carstens.

**Writing – review & editing:** Drew J. Duckett, David C. Tank, Bryan C. Carstens.

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
