## [Decision Letter · Decision Letter 0]

7 Jul 2023

PONE-D-23-05002Reduced representation approaches produce similar results to whole genome sequencing for some common phylogeographic analysesPLOS ONE

Dear Dr. Carstens,

Thank you for submitting your manuscript to PLOS ONE. After careful consideration, we feel that it has merit but does not fully meet PLOS ONE’s publication criteria as it currently stands. Therefore, we invite you to submit a revised version of the manuscript that addresses the points raised during the review process.

Authors must address the comments provided by both reviewers and submit the revised version.

We look forward to receiving your revised manuscript.

Kind regards,

Lalit Kumar Sharma

Academic Editor

PLOS ONE

Journal Requirements:

2. We note that Figure 1 in your submission contain [map/satellite] images which may be copyrighted. All PLOS content is published under the Creative Commons Attribution License (CC BY 4.0), which means that the manuscript, images, and Supporting Information files will be freely available online, and any third party is permitted to access, download, copy, distribute, and use these materials in any way, even commercially, with proper attribution. For these reasons, we cannot publish previously copyrighted maps or satellite images created using proprietary data, such as Google software (Google Maps, Street View, and Earth). For more information, see our copyright guidelines: http://journals.plos.org/plosone/s/licenses-and-copyright.

Natural Earth (public domain): http://www.naturalearthdata.com

Reviewers' comments:

Reviewer's Responses to Questions

**Comments to the Author**

1. Is the manuscript technically sound, and do the data support the conclusions?

Reviewer #1: Yes

Reviewer #2: Yes

2. Has the statistical analysis been performed appropriately and rigorously? 

Reviewer #1: Yes

Reviewer #2: No

3. Have the authors made all data underlying the findings in their manuscript fully available?

Reviewer #1: Yes

Reviewer #2: Yes

4. Is the manuscript presented in an intelligible fashion and written in standard English?

Reviewer #1: Yes

Reviewer #2: No

5. Review Comments to the Author

Reviewer #1: This study is a comprehensive look at various types of genomic sequencing and the return on investment for each using a complicated biological system (North American water vole), as an example. I found it thorough, well written and highly informative. I recommend acceptance without revision.

Reviewer #2: The current quality of the manuscript leaves a number of questions to be answered, which are outlined below. The exposition of the results and its discussion in the manuscript needs a major revision, including careful English usage and style edition.

1. In the ‘Abstract’ section, please take care of the English usage. I would suggest that authors use standard English.

2. The ‘Introduction’ section of the manuscript is not coherent. Also, it does not describe the relevance of the study. A few sentences in the ‘Introduction’ are too long and not making any proper sense. Several colloquial words have been used. Please rewrite the ‘Introduction’ section properly in a concise manner.

3. The outcome of the experiments should be described in the ‘Results’ section, and interpretation in consistence with the previous studies should be given in the ‘Discussion’ section. Accordingly, the results and discussion sections should be revised.

4. Within the text, the citations are not arranged properly; few are alphabetically arranged, while others are chronological. Please maintain a uniform pattern.

5. Please revise the ‘Reference’ section thoroughly according to the format.

Overall, the manuscript presentation is not of adequate quality. The English used in the manuscript is not of standard quality. Many colloquial words have been used, which makes the presentation very poor to be published. I would suggest the authors take the help of an English professional. The manuscript should be revised thoroughly for data presentation, result interpretation, description, and language.

6. PLOS authors have the option to publish the peer review history of their article (what does this mean?). If published, this will include your full peer review and any attached files.

Reviewer #1: No

Reviewer #2: No

---

## [Author Response · Author response to Decision Letter 0]

20 Jul 2023

Reviewer #1: This study is a comprehensive look at various types of genomic sequencing and the return on investment for each using a complicated biological system (North American water vole), as an example. I found it thorough, well written and highly informative. I recommend acceptance without revision.

Duckett et al. (DEA). Thank you for your endorsement of our manuscript. We are pleased that you found it to be well written and informative. While we have conducted a number of edits to the writing, particularly in the Introduction and Discussion, we hope that these have resulted in improvements to clarity and ease of understanding for its readers. 

Reviewer #2: The current quality of the manuscript leaves a number of questions to be answered, which are outlined below. The exposition of the results and its discussion in the manuscript needs a major revision, including careful English usage and style edition.

DEA. Thank you for your review. We have conducted an extensive edit, particularly of the Introduction and Discussion sections, to improve the quality of the presentation throughout the revised manuscript. 

1. In the ‘Abstract’ section, please take care of the English usage. I would suggest that authors use standard English.

DEA. We have rewritten the Abstract to improve clarity. We have attempted to minimize jargon and to use standard English throughout.

2. The ‘Introduction’ section of the manuscript is not coherent. Also, it does not describe the relevance of the study. A few sentences in the ‘Introduction’ are too long and not making any proper sense. Several colloquial words have been used. Please rewrite the ‘Introduction’ section properly in a concise manner.

DEA. We have rewritten the Introduction to improve coherence and to make the relevance of this work more clear. We have replaced many of the words that may have been too colloquial with more technical terms, while endeavoring to maintain readability. 

3. The outcome of the experiments should be described in the ‘Results’ section, and interpretation in consistence with the previous studies should be given in the ‘Discussion’ section. Accordingly, the results and discussion sections should be revised.

DEA. We have revised the Results and Discussion sections to describe the outcomes of the experiments in the Results section and the interpretation of these results in light of previous investigations in the Discussion section. 

4. Within the text, the citations are not arranged properly; few are alphabetically arranged, while others are chronological. Please maintain a uniform pattern.

DEA. The initial submission included references that were formatted alphabetically. It is our understanding that this is encouraged (or at least allowed) by PLoS ONE (e.g., format free initial submission https://journals.plos.org/plosone/s/getting-started ) as a way to minimize the burden on the authors during the initial review process, but we apologize if this is incorrect or if our alphabetically-formated referenced caused undue stress to our reviewers. Regardless, all references are now included in a numerical format following the Vancouver style as described in the author guidelines. 

5. Please revise the ‘Reference’ section thoroughly according to the format.

DEA: All references are now included in a numerical format following the Vancouver style as described in the author guidelines.

Overall, the manuscript presentation is not of adequate quality. The English used in the manuscript is not of standard quality. Many colloquial words have been used, which makes the presentation very poor to be published. I would suggest the authors take the help of an English professional. The manuscript should be revised thoroughly for data presentation, result interpretation, description, and language.

DEA: We have taken this opportunity to thoroughly edit this manuscript for clarity and language with the goal of improving the data presentation and interpretation of results. Thank you for your review of our work.

---

## [Editor Report · Decision Letter 1]

10 Sep 2023

Reduced representation approaches produce similar results to whole genome sequencing for some common phylogeographic analyses

PONE-D-23-05002R1

Dear Dr. Carstens,

We’re pleased to inform you that your manuscript has been judged scientifically suitable for publication and will be formally accepted for publication once it meets all outstanding technical requirements.

Kind regards,

Lalit Kumar Sharma

Academic Editor

PLOS ONE
---

## [Editor Report · Acceptance letter]

17 Nov 2023

PONE-D-23-05002R1 

Reduced representation approaches produce similar results to whole genome sequencing for some common phylogeographic analyses 

Dear Dr. Carstens:

I'm pleased to inform you that your manuscript has been deemed suitable for publication in PLOS ONE. Congratulations! Your manuscript is now with our production department. 

Kind regards, 

on behalf of

Dr. Lalit Kumar Sharma 

Academic Editor

PLOS ONE